# An Investigation on the Sound Absorption Performance of Granular Molecular Sieves under Room Temperature and Pressure

**DOI:** 10.3390/ma13081936

**Published:** 2020-04-20

**Authors:** Bing Zhou, Jiangong Zhang, Xin Li, Bilong Liu

**Affiliations:** 1State Key Laboratory of Power Grid Environmental Protection, China Electric Power Research Institute, Wuhan 430074, China; zhoubing2@epri.sgcc.com.cn; 2School of Mechanical and Automobile Engineering, Qingdao University of Technology, Qingdao 266520, China; jz03-4lx@163.com

**Keywords:** molecular sieve pellets, sound absorption, impedance tube

## Abstract

The sound absorption of granular silica-aluminate molecular sieve pellets was investigated in this paper. The absorption coefficients of molecular sieve pellets with different pore sizes, pellet sizes, and layer thicknesses were measured through impedance tubes under room temperature and pressure conditions. The effects of pore size, pellet size, layer thickness were compared and explained. The comparisons show that at room temperature and pressure, the sound absorption of molecular sieve pellets is not a result of the crystalline structure, but rather it mainly changes with the pellet size and layer thickness. In addition, the five non-acoustical parameters of molecular sieve pellets were obtained by an inverse characterization method based on impedance tube measurements. The measurement by impedance tubes is in good agreement with the calculation of Johnson-Champoux-Allard (JCA) model, proving that the JCA model can be effectively used to predict the sound absorption of molecular sieve pellets.

## 1. Introduction

Porous materials are the most widely used sound absorbing materials in engineering. The obvious sound absorption of porous materials lies in the fact that their internal structure has numerous tiny pores that are interlinked with each other and connected to the outside on their surfaces. During the process of sound wave propagation, friction occurs between the moving air in the pores and the solid skeleton of the pore wall, and viscous loss is caused by the viscosity of airflow within the materials. Due to the viscosity and heat conduction effect of the air, sound energy is converted into internal energy and then dissipated, thus achieving sound absorption [1].

Porous sound absorbing materials may be classified into fibrous, cellular and granular [2,3]. Granular materials are considered as rigid porous materials due to their fixed framework, and their sound energy dissipation mainly depends on the air viscosity in the gaps of the granular materials and the heat transfer between the air and the granular material surface. Meanwhile, the scattering of granules will also affect the absorption of sound energy in the material. References [4,5,6,7,8,9,10] studied the sound absorption of some granular porous materials such as rubber chips, gravel, perlite, aerogel and concrete. The results show that these granular porous materials play a role in sound absorption and noise reduction of road noise barriers and buildings. Molecular sieves are a kind of granular porous material that have been widely used as air adsorbents and desiccants in many applications [11,12] Quite recently, granular molecular sieves have been used as sound absorption materials inside the back cavities of miniature loudspeakers for cellphones [13,14]. It is considered that the air absorption mechanism increases the acoustic compliance of a volume of molecular sieves, which significantly reduces the first resonance frequency of the coupled membrane-cavity system and thus improves the sound performance of loudspeakers. For underwater acoustics, a damping material consisting of Rho c-carbon fiber matrix and molecular sieve pellets was reported, and experimental results showed that the addition of molecular sieve pellets improved the sound insulation [15]. On the topic of the sound absorption characteristics of molecular sieves as porous materials, however, there are few reports making this worth investigating in some detail.

Molecular sieves, also known as zeolites, belong to the aluminosilicate compounds, a sort of crystalline porous material. The corresponding crystallographic structure is formed by tetrahedrons of AlO_4_ and SiO_4_, which are the basic components of different zeolites. The crystalline structure of molecular sieves is a three-dimensional porous system that precisely adjusts the pore size to allow molecules smaller than this pore size to be adsorbed and to exclude larger molecules [11,12], hence the name “molecular sieve”. Molecular sieves are usually divided into A, X, Y, M and ZSM types according to their ratio of aluminum to silicon. Generally speaking, the higher the Al-Si ratio of a zeolite is, the better its stability is. For example, the initial damage temperature of the structure of NaA (SiO_2_/Al_2_O_3_ = 2) and NaX (SiO_2_/Al_2_O_3_ = 2.5) is approximately 660 °C. Molecular sieves with different crystal structures are usually classified by the pore diameter, such as 3Å, 4Å and 5Å, where 1Å = 0.1 nm. 3Å molecular sieves have effective adsorption selectivity and are suitable for the insulating glass desiccant and refrigerant desiccant applications; 5Å molecular sieves have high adsorption and fast adsorption speed, and are suitable for various gas pressure swing adsorption devices for oxygen, hydrogen and carbon dioxide production. The appearance of molecular sieves is usually as a powder, strips or granular. The size range of molecular sieve pellets is generally 0.4–0.8, 1.6–2.5 or 3–5 mm. For convenience, we will uniformly refer to them as micro, small and large pellets in this paper.

The acoustic properties of porous materials are mainly focused on the propagation of sound waves through their interior. Zwikker and Kosten [16] pointed out that for isotropic homogeneous media, the propagation of sound waves can be described by the characteristic impedance and propagation constant of the materials. Due to the complexity of the factors affecting the sound absorption of porous materials, empirical models are generally used to estimate the sound absorption performance. Delany-Bazley [17] obtained an empirical formula to describe the characteristic impedance and wave number of porous materials through a large number of measurements on fibrous materials. This empirical model is very successful since it requires only the static flow resistivity of materials. The model, however, is not applicable to the prediction of sound absorption in wide frequency band. Miki [18] further modified the above formula to make it applicable to a wider frequency range. In addition to the empirical formula above, many scholars proposed a semi-phenomenological model to describe the equivalent dynamic density and equivalent volume modulus of fluids in rigid skeletal porous materials [19,20,21,22,23,24,25]. In 1987, Johnson et al. [19] proposed a semi-phenomenological model to describe the complex density of porous materials with arbitrary pore shapes and static skeletons. Taking into account the viscosity-inertia effect, four parameters that are static flow resistivity, porosity, tortuosity and viscosity characteristic length, were used to calculate dynamic density. In 1991, Champoux and Allard [20] further considered the heat conduction effect and gave the expression of dynamic volume modulus of porous materials with the same property. Five parameters, which are static flow resistivity, porosity, tortuosity, viscous characteristic length and thermal characteristic length, were used to calculate the equivalent fluid dynamic density and equivalent fluid dynamic volume modulus of the porous material model. Johnson-Champoux-Allard (JCA) model is a generalized model suitable for the wide-band propagation of sound waves in porous materials [21,22,23] Non-acoustic parameters of porous materials can be obtained through direct measurement or inverse acoustical characterization [26,27]. Direct measurement requires special laboratory equipment. Static flow resistance and porosity can be directly measured by standard technology, while tortuosity and viscous characteristic length, thermal characteristic length can be measured by ultrasonic technology, which is difficult to implement under conventional laboratory conditions, and the measurement accuracy cannot easily be guaranteed. Inverse acoustic characterization is an alternative to direct measurements. Once the acoustic impedance or absorption coefficient of the material is known, the relevant parameters of the material can be calculated using the inverse strategy. In [7,26,28,29] the non-acoustic parameters of porous materials are evaluated using inverse acoustic characterization, and the results show that this method is robust and reliable. Here, taking the minimum difference between the sound absorption coefficient measured by the impedance tube and calculated by the JCA model as the objective function, five non-acoustic parameters of molecular sieve are obtained by using simulated annealing method.

The paper is organized as follows: In Section 2, sound absorption measurements of different molecular sieve pellets samples with different thickness and different pellet size are reported and analyzed; and in Section 3, JCA model is adopted to describe the acoustical properties of molecular sieve pellets. Five non-acoustic parameters of the molecular sieve pellets are calculated by simulated annealing. Moreover, the reliability of JCA model for predicting the sound absorption of molecular sieve pellets is verified. Finally, the conclusions are summarized in Section 4.

## 2. Acoustic Measurement of Molecular Sieve Pellets

### 2.1. Experimental Equipment and Samples

The purpose of this paper is to investigate the sound absorption characteristics of molecular sieves at room temperature and pressure. 3Å and 5Å molecular sieve pellets were chosen for the measurements carried out in SW impedance tubes according to ISO 10534-2 [30]. Based on the transfer function method, the normal incident sound absorption coefficient and impedance in the 63–6300 Hz range are measured by SW422 and SW477 impedance tubes, of which the SW422 has an inner diameter of 100 mm and is used for 63–1600 Hz and the SW477 has 30 mm and is used for 1000–6300 Hz measurements. As for the impedance tube test system in Figure 1, a BSWA PA50 power amplifier drives the speaker to amplify the sound, the microphones located at positions 1 and 2 collect the sound pressure of the incident and reflected waves, and a MC3242 data acquisition system analyses the sound pressure signals. The reflection factor of the sound waves is calculated by the transfer function of incident and reflected waves in sound field. Then the normal incident coefficient and surface acoustic impedance are calculated. The surface acoustic impedance and impedance are the real and imaginary parts of the surface acoustic impedance, respectively.

A perforated cylindrical box is designed to fill with loose molecular sieve pellets. The sound absorption of the perforated cylinder is very low and can be ignored here. For example, a perforated cylinder with an aperture of 1.5 mm and a perforation ratio of 39.6% is used, and its measurement of sound absorption coefficient with cylinder depths of 20, 30 and 50mm is shown on the right of Figure 1. It should be noted that, in the stacking state, the void ratio of bulk granular materials is described by the packing density and apparent density and is related to the sound absorption of granular materials. The void ratio is inversely proportional to the packing density of materials. Molecular sieves pellets described here are randomly closely packed in the specimen cylinder in a natural state without additional force or adhesive compaction. In the following sections, the measured normal sound absorption coefficient and surface acoustic impedance of molecular sieve pellets of different pore size and different pellet sizes, and different thickness are given and compared.

### 2.2. Measured Sound Absorption of Molecular Sieve Pellets with Different Pore Size

In order to find out the effect of pore size on the sound absorption of molecular sieve pellets under room temperature and pressure, two types of samples with pore sizes of 3Å and 5Å are selected for sound absorption comparison. The characteristics of the molecular sieve pellet samples used are listed in Table 1. Figure 2a–d show the sound absorption curves of molecular sieve pellets of the same layer thick and pellet size.

The sound absorption curves of 3Å and 5Å are similar in shape and almost identical. The slight differences in some frequency bands may be caused by installation or measurement errors. Under room temperature and pressure, “sieve molecules” does not appear due to its own strengthening stability. That is to say, the pore size has no obvious effect on the sound absorption of molecular sieve pellets. The sound absorption of molecular sieve pellets is mainly related to the macroscopic size of pellets and is less related to the pore size. Therefore, molecular sieve pellets are regarded as rigid porous materials, and sound energy is mainly consumed by the air viscosity between pellets and heat transfer between air and pellets surface. In addition, a low sound absorption coefficient valley appears on each sound absorption curve, which obviously affects the wide-band sound absorption of molecular sieve pellets. This is because the sound absorption of molecular sieve pellets is related to flow resistance, porosity and structure factor, and finally attributed to the pellets gap. When the gap between pellets is too large or too small, the air between the pellets cannot be sufficiently frictional and heat transfer during the propagation of sound waves.

### 2.3. Measured Sound Absorption of Molecular Sieve Pellets with Different Thickness

To illustrate the effect of layer thickness on the sound absorption of molecular sieve pellets, the measured sound absorption coefficient for the eight samples given in Table 2 are plotted in Figure 3. As shown in Figure 3a,b, as the thickness of the molecular sieve pellets increases, the absorption peak tends to the low frequency and as well the effective absorption bandwidth becomes narrower. As the thickness of the layer increases, the gap distribution becomes more complex and the gap channel becomes longer. In this way, during the acoustic wave propagation process, the number of collisions between pellets increases and more energy is consumed, and the flow volume in the gap increases, so the natural frequency of molecular sieve pellets decreases and the sound absorption moves to a low frequency.

### 2.4. Measured Sound Absorption of Molecular Sieve Pellets with Different Pellet Sizes

Three different pellet size molecular sieve pellets selected for sound absorption comparison are illustrated in Figure 4, where the (a1) and (b1) are the sound absorption coefficient of the samples, (a2) and (b2) are the specific surface acoustic resistance of the samples, and (a3) and (b3) are the specific surface acoustic resistance of the samples. The pellet sizes are micro (0.4–0.8 mm), small (1.6–2.5 mm), and large (3–5 mm), as shown in Table 3. It is evident that as the pellet size decreases, the sound absorption peaks of the molecular sieve pellets move to a low frequency, and the effective sound absorption bandwidth increases significantly. Compared with the large and small samples, the sound absorption coefficient of the micro sample is smaller and flat, as shown in (a1) and (b1). The sound absorption coefficient of the micro sample is greater than 0.6 in the bandwidth of 500–6300 Hz. When the molecular sieve pellets are smaller, the pellet gap is smaller and denser, and the fluid resistance in the gap increases, thus more sound energy is lost. This characteristic is clearly reflected from the measured specific surface acoustic resistances in (a2) and (b2), where the micro sample has the largest acoustic resistance excluding the frequency at resonance. Also the measured specific surface acoustic reactance of the micro sample are the nearest zero in the whole frequency range, therefore the smallest samples have the benefit for broadband sound absorption. These results imply that appropriate pellet size should be taken into account carefully to have a satisfactory sound absorption.

## 3. JCA model and Inverse Parameters Estimation

### 3.1. Theoretical Model

JCA model is a generalized model for sound propagation in porous materials over a wide range of frequencies, which includes five non-acoustic parameters of porous materials: the static flow resistance σ, porosity ϕ, the tortuosity $\alpha_{\infty }$, the viscous characteristic length Λ, and the thermal characteristic length Λ′. In JCA model, the equivalent dynamic density $\rho_{e}\left(\omega \right)$ and the equivalent dynamic volume modulus $K_{e}\left(\omega \right)$ are expressed as [20]:(1)$$\rho_{e}\left(\omega \right)=\rho_{0}\alpha_{\infty }\left(1+\frac{\phi \sigma }{j\omega \rho_{0}\alpha_{\infty }}{\left(1+j\frac{4\omega \rho_{0}\eta \alpha_{\infty }{}^{2}}{{\left(\sigma \phi \Lambda \right)}^{2}}\right)}^{1/2}\right)$$
(2)$$K_{e}\left(\omega \right)=\frac{\gamma P_{0}}{\gamma -\left(\gamma -1\right){\left(1+\frac{8\eta }{j\omega B^{2}\Lambda \prime^{2}\rho_{0}}{\left(1+j\frac{\omega B^{2}\rho_{0}\Lambda \prime^{2}}{16\eta }\right)}^{1/2}\right)}^{-1}}$$
(3)$$\Lambda =\frac{1}{c}{\left[\frac{8\alpha_{\infty }\eta }{\sigma \phi }\right]}^{1/2}$$
(4)$$\Lambda \prime =\frac{1}{c\prime }{\left[\frac{8\alpha_{\infty }\eta }{\sigma \phi }\right]}^{1/2},$$
where *P_0_* is the ambient atmospheric pressure, ω is the angular frequency, $\rho_{0}$, η, γ and $B^{2}$ are the density, dynamic viscosity, specific heat ratio and Prandtl number of the saturating air, respectively. In Equations (3) and (4), c and c′ are pore shape parameters related to the viscous and thermal dissipation, respectively.

Based on the equivalent fluid theory, the characteristic acoustic impedance $Z_{c}$ and complex wave number k of rigid porous materials are expressed as:(5)$$Z_{c}=\frac{1}{\phi }\sqrt{\rho_{e}\left(\omega \right)gK_{e}\left(\omega \right)}$$
(6)$$k=\omega \sqrt{\frac{\rho_{e}}{K_{e}}}.$$

Therefore, according to Zwicker and Kosten theory, the surface acoustic impedance $Z_{s}$ of porous materials backed by rigid walls is expressed as:(7)$$Z_{s}=Z_{c}\cot h(kL).$$

Then, the normal incidence sound absorption coefficient can be calculated as:(8)$$\alpha =1-{\left|\frac{Z_{s}-\rho_{0}c_{0}}{Z_{s}+\rho_{0}c_{0}}\right|}^{2},$$
where, L is the thickness of the porous material layer, $\rho_{0}c_{0}$ is the characteristic impedance of the air.

### 3.2. Inversion of Non-Acoustic Parameters

Simulated annealing is an optimization algorithm based on Monte Carlo iteration. Because of its global optimization and strong robustness, it is widely used in the optimization of multi-objective parameters in acoustic materials and structures [26,31,32,33,34]. In this paper, the inverse calculation of five non-acoustic parameters is transformed into the solution of a set of optimal values. The process is as follows: firstly, the difference between the sound absorption coefficient measured in the impedance tube and calculated by JCA model in a certain frequency band is taken as the optimization objective function, and then a set of optimization values are obtained by global optimization calculation using simulated annealing method. The objective function is as follows:(9)$$F\left(\alpha \right)=\sum_{i}|\alpha_{Measured}\left(f_{i}\right)-\alpha_{JCA}\left(f_{i}\right)|,$$where $f_{i}$ is the frequency point within the specified frequency range, $\alpha_{Measured}\left(f_{i}\right)$ is the sound absorption coefficient at the frequency point $f_{i}$ measured by the impedance tube, $\alpha_{JCA}\left(f_{i}\right)$ is the sound absorption coefficient calculated by JCA model.

Meanwhile, according to the analysis of five non-acoustic parameters in literature [26], the optimization interval of five parameters in this paper is set as:(10)$$\{\begin{array}{l} \sigma \in \left[\begin{array}{cc} 1000 & 10,000 \end{array}\right]Pa\cdot s\cdot m^{-2}\\ \phi \in \left[\begin{array}{cc} 0.1 & 0.9 \end{array}\right]\\ \alpha_{\infty }\in \left[\begin{array}{cc} 1 & 4 \end{array}\right]\\ c\in \left[\begin{array}{cc} 0.3 & 3.3 \end{array}\right]\\ c\prime \in \left[\begin{array}{cc} 0.3 & 3 \end{array}\right] \end{array}.$$

Using the above inversion strategy, the calculated non-acoustic parameters of samples are listed in Table 4. The sound absorption coefficient measured using impedance tubes and estimated by substituting the inversion values into the JCA model are respectively drawn in Figure 5. It can be found that in the frequency range of 63-6300Hz, the measured curve is almost identical to the estimated curve.

### 3.3. Validation

New samples with different thickness listed in Table 5 are chosen to verify that the inversion parameters estimated in Table 4. The sound absorption coefficients of these new samples measured and predicted using the inversion parameters are shown in Figure 6. The prediction of sample #1 and #2 is based on the inversion parameters of sample 3Å in Table 4 and the sample #3 and #4 is according to sample 5Å in Table 4. It is evident that there is good agreement between the predicted and measured sound absorption coefficients for these samples. Therefore, the inversion parameters appear to be independent of the thickness of molecular sieve pellets. This implies that the JCA model can be used as a reliable model to predict the sound absorption performance of molecular sieve pellets, and the five non-acoustic parameters obtained from one sample by inverse method based on the JCA model and simulated annealing are reliable to be used for the prediction of the sound absorption coefficient of molecular sieve pellets with different thickness.

## 4. Conclusions

Under the conditions of room temperature and pressure, the sound absorption performance of molecular sieve pellets was measured for comparisons. It is found that the sound absorption coefficients are very sensitive to the pellet size and the thickness of the molecular sieve pellets, but not sensitive to the pore size of the molecular sieve pellets, implying that the sound absorption is not caused by the crystalline structure. Molecular sieve pellets with optimized pellet size may result in satisfactory absorption in the broad band frequency range. Increasing the thickness of the molecular sieve pellets moves the sound absorption peaks to a lower frequency range. Moreover, it has been demonstrated that the JCA model is appropriate to describe the sound absorption performance of molecular sieve pellets. The five non-acoustic parameters obtained with one sample by the inverse method are reliable and can be used for the prediction of the sound absorption coefficient of molecular sieve pellets with different thicknesses. When the temperature and pressure are higher than those of room conditions, or under the condition of high sound intensity, the effect of the crystalline structure on the sound absorption has not been verified and is worth being investigated in the future.

## Figures and Tables

**Figure 1 materials-13-01936-f001:**
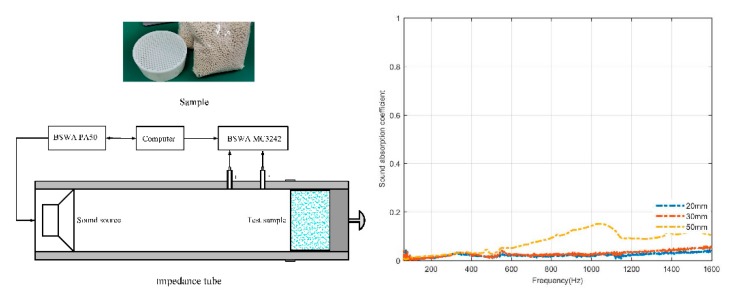
Experiment setup. On the left are the sample and the impedance tube, and on the right is the absorption coefficient of the cylindrical box.

**Figure 2 materials-13-01936-f002:**
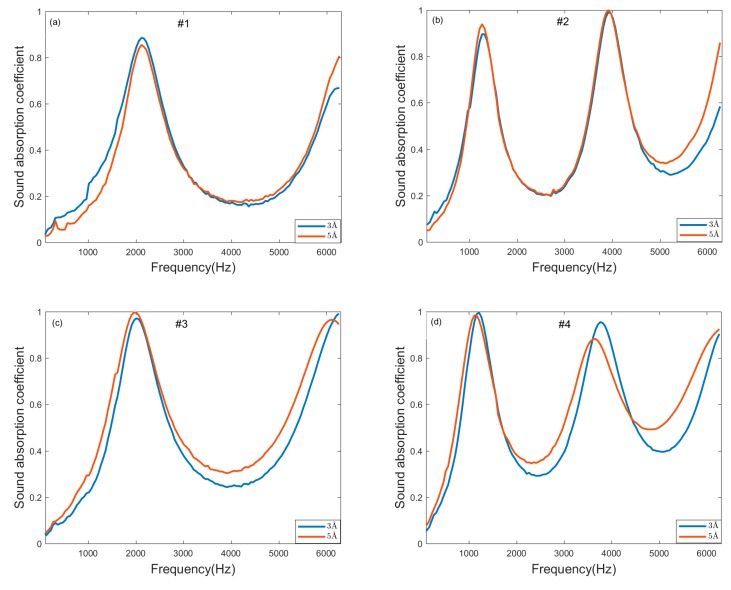
Sound absorption measurement of molecular sieve pellets of different pore size. Sound absorption coefficient of 3Å and 5Å with (**a**) thickness of 30 mm and pellet size of 3–5 mm; (**b**) thickness of 50 mm and pellet size of 3–5 mm; (**c**) thickness of 30 mm and pellet size of 1.6–2.5 mm; (**d**) thickness of 50 mm and pellet size of 1.6–2.5 mm.

**Figure 3 materials-13-01936-f003:**
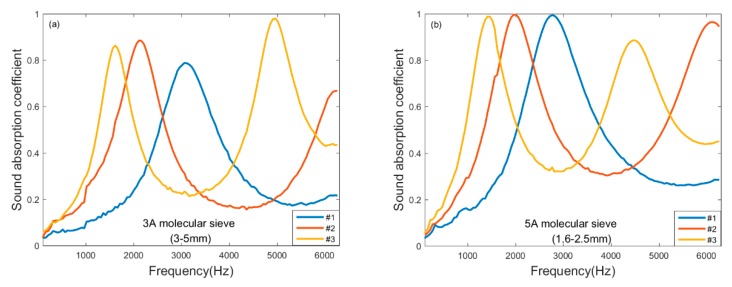
Sound absorption measurement of molecular sieve pellets with different thickness: (**a**) 3Å with pellet size of 3–5 mm; (**b**) 5Å with pellet size of 1.6–2.5 mm.

**Figure 4 materials-13-01936-f004:**
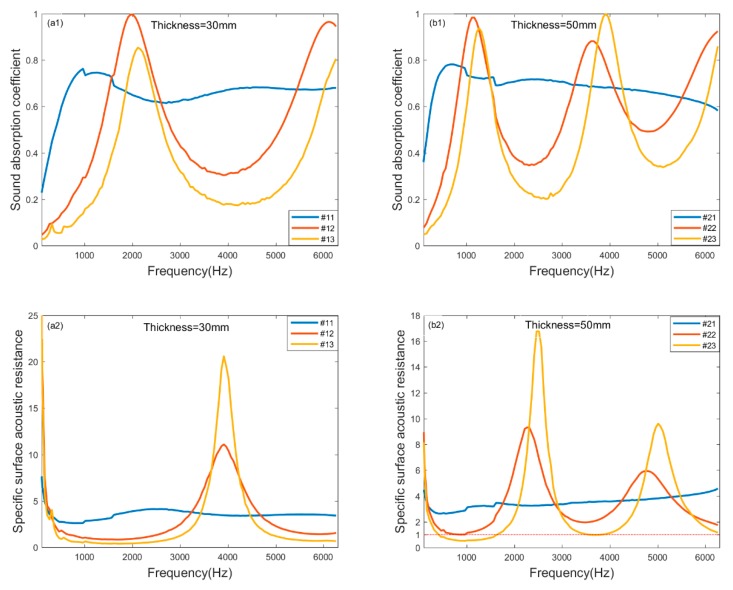
Sound absorption measurement of molecular sieve pellets with different pellet sizes. Sound absorption coefficients of 5Å with thickness of (**a1**) 30 mm; (**b1**) 50 mm; Specific surface acoustic resistance of 5Å with thickness of (**a2**) 30 mm; (**b2**) 50 mm; Specific surface acoustic reactance of 5Å with thickness of (**a3**) 30 mm; (**b3**) 50 mm.

**Figure 5 materials-13-01936-f005:**
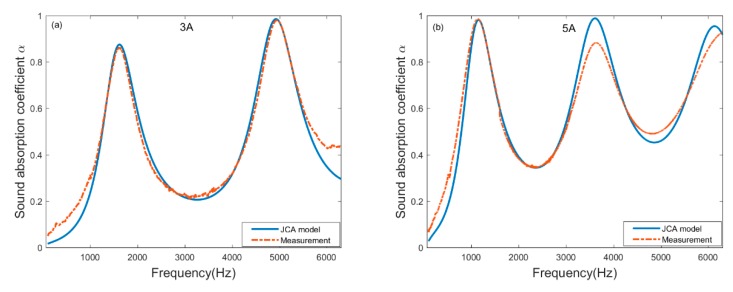
Comparison between measurement by impedance tube and estimation through inversion of JCA model. (**a**) 3Å molecular sieve pellets with thickness of 40 mm; (**b**) 5Å molecular sieve pellets with thickness of 50 mm.

**Figure 6 materials-13-01936-f006:**
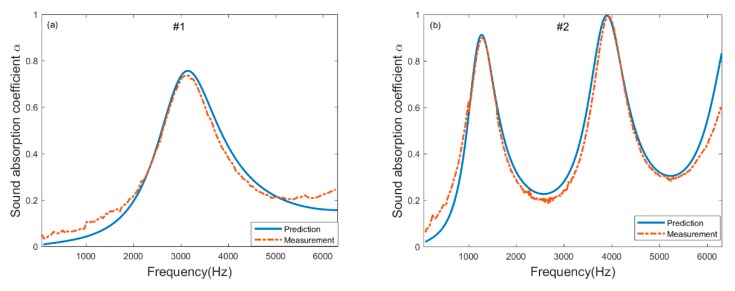
Comparison between measurement and prediction for the samples listed in using the inversion parameters estimated in **(a)** new sample #1**; (b)** new sample #2; **(c)** new sample #3; **(d)** new sample #4.

**Table 1 materials-13-01936-t001:** Characteristics of the molecular sieve pellet samples with different pore sizes.

3Å			5Å		
Number	Pellet Size (mm)	Thickness (mm)	Number	Pellet Size (mm)	Thickness (mm)
#1	3–5	30	#1	3–5	30
#2	3–5	50	#2	3–5	50
#3	1.6–2.5	30	#3	1.6–2.5	30
#4	1.6–2.5	50	#4	1.6–2.5	50

**Table 2 materials-13-01936-t002:** Samples of molecular sieve pellets with different layer thickness.

3Å			5Å		
Number	Pellet Size (mm)	Thickness (mm)	Number	Pellet Size (mm)	Thickness (mm)
#1	3–5	20	#1	1.6–2.5	20
#2	3–5	30	#2	1.6–2.5	30
#3	3–5	40	#3	1.6–2.5	40

**Table 3 materials-13-01936-t003:** Samples of molecular sieve pellets with different pellet sizes.

5Å					
Number	Pellet Size (mm)	Thickness (mm)	Number	Pellet Size (mm)	Thickness (mm)
#11	0.4–0.8	30	#21	0.4–0.8	50
#12	1.6–2.5	30	#22	1.6–2.5	50
#13	3–5	30	#23	3–5	50

**Table 4 materials-13-01936-t004:** Inverse non-acoustic parameters estimated by simulated annealing.

Sample	Pellet Size (mm)	Thickness (mm)	σ (Pa·s·m^−2^)	ϕ (%)	*α_∞_*	Λ (μm)	Λ′ (μm)	c	c′
3Å	3–5	40	4535	39.39	1.62	360.32	540.77	1	0.66
5Å	1.6–2.5	50	4503	44.62	1.68	216.20	345.93	1.60	1

**Table 5 materials-13-01936-t005:** Samples of molecular sieve pellets for prediction.

3Å			5Å		
Number	Pellet Size (mm)	Thickness (mm)	Number	Pellet Size (mm)	Thickness (mm)
#1	3–5	20	#3	1.6–2.5	30
#2	3–5	50	#4	1.6–2.5	40

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
